# Can Vitality and Mental Health Influence Upper Extremity Pain? A Prospective Cohort Study of 1185 Female Hospital Nurses

Rodrigo Núñez-Cortés [1,2,3], Ander Espin [1,4,5], Joaquín Calatayud [1,6,*], Sofía Pérez-Alenda [2],
Carlos Cruz-Montecinos [3,7], Rubén López-Bueno [1,5,8], Jonas Vinstrup [1], Markus D. Jakobsen [1]
and Lars Louis Andersen [1]

1 National Research Centre for the Working Environment, 2100 Copenhagen, Denmark;
r_nunez@uchile.cl (R.N.-C.); ander.espin@ehu.eus (A.E.); rlopezbu@unizar.es (R.L.-B.); jov@nfa.dk (J.V.);
mdj@nfa.dk (M.D.J.); lla@nfa.dk (L.L.A.)
2 Physiotherapy in Motion Multispeciality Research Group (PTinMOTION), Department of Physiotherapy,
University of Valencia, 46010 Valencia, Spain; sofia.perez-alenda@uv.es
3 Department of Physical Therapy, Faculty of Medicine, University of Chile, Santiago 8370000, Chile;
carloscruz@uchile.cl
4 Ageing on Research Group, Department of Physiology, University of the Basque Country (UPV/EHU),
48940 Leioa, Spain
5 Biocruces Bizkaia Health Research Institute, 48903 Barakaldo, Spain
6 Exercise Intervention for Health Research Group (EXINH-RG), Department of Physiotherapy, University of
Valencia, 46011 Valencia, Spain
7 Division of Research, Development and Innovation in Kinesiology, Kinesiology Unit, San José Hospital,
Santiago 8370000, Chile
8 Department of Physical Medicine and Nursing, University of Zaragoza, 50009 Zaragoza, Spain
* Correspondence: joaquin.calatayud@uv.es

**Abstract:** Musculoskeletal disorders of the upper extremity are among the most common occupational problems affecting nurses. The aim of this study was to analyze the prospective association between vitality and mental health and increased upper extremity pain intensity in female hospital nurses during a 1-year follow-up. A prospective cohort of 1185 female nurses from 19 hospitals in Denmark was conducted using baseline and 12-month follow-up questionnaires to identify potential associations between levels of vitality and mental health (SF-36 subscales) with pain intensity (0–10 scale) in the shoulder, elbow and hand/wrist regions. Associations were modeled using cumulative logistic regression. The fully adjusted model included the variables of age, baseline pain, body mass index, smoking status, years of occupation, leisure time physical activity level, number of daily patient transfers/handlings, as well as recognition and influence at work. The mean age was 48.3 (SD: 10.4) years. In the fully adjusted model, significant associations between low vitality levels and the odds of shoulder pain (OR = 1.96; 95%CI: 1.43–2.68) and hand/wrist pain (OR = 2.32; 95%CI: 1.58–3.42) were observed. Likewise, moderate levels of mental health was associated with increased odds of shoulder pain at follow-up (OR = 1.50; 95%CI: 1.16–1.93). These results provide an important incentive for nursing managers to assess vitality and mental health among hospital nurses and to consider this factor in prevention strategies to ensure good worker health and, by extension, high-quality care.

**Keywords:** pain; nursing care; upper limb; psychosocial functioning; emotions; woman health

## 1. Introduction

Musculoskeletal disorders in the upper extremity are among the most common work-related problems affecting nurses [1]. Nursing is a physically demanding job involving manual patient handling, awkward working positions, and forceful movements of the

upper extremities (e.g., changing patient dressings, injections, sterile washing of patients, and patient transfers) [2–4]. Thus, nurses exhibit high prevalences of musculoskeletal disorders [5,6], with numbers being significantly higher among female nurses compared to their male counterparts [7]. Previous research has suggested that female workers may experience higher levels of upper limb pain compared to male workers [8]. Women's higher risk of injury, particularly in the upper limbs, may be explained by biological differences such as lean muscle mass and the endocrine system, and cultural gender stereotypes where women tend to do more repetitive and monotonous work than men [9,10].

A recent meta-analysis investigated the prevalence of work-related musculoskeletal disorders among hospital nurses, reporting concerningly high values for the shoulder (44%), elbow (18%), and hand/wrist (29%) regions [11]. Also, several studies have indicated that the shoulder is one of the most affected regions among this subgroup of the working population [1,7,12], with reported prevalences as high as 85.8% [12]. Likewise, although having received less attention in clinical research, hand/wrist pain is also common in hospital nurses, and its pathology is associated with multiple factors, including repetitive wrist or finger movements, high work stress, and pain catastrophizing, as well as mental and physical well-being [13]. In this context, musculoskeletal disorders may result in reduced work capacity, increased staff turnover, and a negative impact on the quality of patient care, which, in turn, increases the collective burden on employers, employees, and society [12,14,15]. However, in addition to the physically demanding tasks that are inherently present within the field of healthcare, nurses also experience high levels of job stress, high job demands, poor job control, and poor social support [16–18]. In fact, a recent meta-analysis identified convincing associations between upper extremity musculoskeletal pain and a wide range of work-related psychosocial factors, including tedious work, inadequate staffing, job demands, insufficient support, time pressure, decision latitude, job dissatisfaction, and job stress [19]. However, while work-related risk factors are somewhat firmly established, investigations focusing on the relationship between psychosocial variables specific to the individual healthcare worker are lacking. For example, potential associations between vitality, mental health, and upper extremity pain in nurses have been scarcely explored in the literature.

Vitality is, most commonly, defined as a unique, unified construct, distinguishable from that of depression and anxiety [20]. Due to the subjective feeling of high energy and low fatigue, higher levels of vitality could be a protective factor against excessive job strain. In fact, vitality has previously been associated with employee performance and well-being [21,22], and high vitality likely allows for greater emotional and social functioning, as well as being associated with better mental health [23]. A recent study of participants in five European countries (France, Germany, Italy, Spain, and the UK) found that people with low vitality are more likely to have problems with mobility, pain/discomfort, depression and anxiety, and impaired usual activity [24]. In addition, people who did not engage in the self-management of their health were more likely to have low vitality [24]. Vitality levels are therefore likely to influence how health professionals perceive stressful situations [20]. In this sense, positive coping strategies are associated with greater well-being and better quality of work life [25], which could indirectly result in better decision making and higher quality care for their patients. Thus, investigation into these prognostic factors—and their relationship with musculoskeletal disorders in nurses—may be useful in convincing managers of the importance of prioritizing vitality and mental health.

On the other hand, a recent meta-analysis showed a statistically significant correlation between mental health factors (i.e., symptoms of anxiety and depression) and absenteeism in people with upper limb disorders [26]. Both absenteeism and presenteeism (loss of productivity due to being at work while ill and underperforming) result in a significant economic burden for workers with mental health and musculoskeletal conditions [27]. In this regard, it is important to identify those cases that may present a higher risk of increased upper limb pain in nurses working in hospital patient transfer units, where the physical burden is greater. The objective was to analyze the prospective association between

baseline vitality and mental health and increased upper extremity pain intensity in female hospital nurses during a 1-year follow-up. It was hypothesized that poorer vitality and mental health scores would be associated with a higher risk of increased upper extremity pain intensity.

## 2. Materials and Methods

### 2.1. Study Design and Participants

A prospective cohort study with baseline and 12-month follow-up questionnaires was conducted to identify associations between vitality and mental health levels with pain intensity in the shoulder, elbow, and hand/wrist regions. Baseline questionnaires were sent by e-mail to 19 public hospitals in Denmark, representing the Northern and Central region of the country. Only female nurses were included. Nursing departments where the physical transfer of patients took place were included, so administrative and clerical departments were excluded. A total of 1185 hospital female nurses completed the entire questionnaire (i.e., at baseline and 1-year follow-up) and were included in this study (59% response rate). The mean age was 48.34 (SD: 10.38) years. The National Research Centre for the Working Environment has an agreement with the Danish Data Protection Agency to register all studies in-house. According to Danish legislation, studies based on research questionnaires and registers do not require approval from ethical and scientific committees or informed consent (Act on Scientific Ethical Committees, §14.2) [28]. All data were de-identified and analyzed anonymously. This article conforms to the Strengthening the Reporting of Observational Studies in Epidemiology (STROBE) statement [29].

### 2.2. Predictor Variables

At baseline and the 12-month follow-up, participants completed the Danish 36-item Short Form Health Survey (SF-36) [30]. Vitality was determined using a three-question subscale of the SF-36, which asked "To what extent during the past four weeks: (a) did you feel full of pep?; (b) have a lot of energy?; (c) feel exhausted?" Participants responded on a 6-point scale from 1—"All the time" to 6—"Not at all". Mental health was determined by the four-question subscale of the SF-36, which asked "How much of the time during the past four weeks: (a) have you been a very nervous?; (b) felt so depressed that nothing could cheer you up?; (c) felt calm and peaceful?; (d) felt depressed and sad?" Participants responded on a 6-point scale from 1—"All the time" to 6—"Not at all". Each dimension was scored by coding, summing, and transforming the item scores according to the scoring guidelines [31], ranging from 0 (worst possible health state) to 100 (best possible health state). For the analyses, both scales were classified into three levels, i.e., low (0 to 50), moderate (>50 to 75), and high (>75 to 100). These cut-off points were taken into account considering that the components of the SF-36 questionnaire are standardized to reflect a mean of 50 in the general population. Whereas scores above 75 have been described for the young population in the European population, weighted by age and sex [32].

### 2.3. Main Outcome

The outcome of the present study was a 1-point increase in pain intensity from baseline to 12 months, assessed using a numerical scale from 0 to 10, with 0 being no pain and 10 being the worst pain imaginable [33]. Participants were asked to rate their average pain during the previous four weeks in the shoulder, elbow, and hand/wrist regions. We chose an increase of one point on the pain scale because this is the smallest possible increase on this ordinal scale. It was also taken into account that the baseline pain intensity in the sample was low. In this sense, an increase of one point could be relevant, as a previous study showed that even low levels of pain affect functioning in people with chronic pain [34].

### 2.4. Covariates

The following potential confounders were included in subsequent analyses: age, pain intensity at baseline, body mass index (BMI), smoking status (yes/no), years in profession,

level of leisure-time physical activity (sedentary; light exercise >3/week; moderate exercise >3/week; vigorous exercise several times per week), and frequency of daily patient transfers. Finally, recognition and influence at work, based on the Copenhagen Psychosocial Questionnaire [35], were also included.

*2.5. Data Analysis*

All data analyses were performed with SAS version 9.4 (Proc Genmod, SAS Institute, Cary, NC, USA). Associations between vitality and mental health (predictor variables at baseline) and pain intensity (outcome) were modeled using cumulative logistic regression, expressing the odds of pain increasing by one point on the 0–10 scale during follow-up. Estimates are provided as odds ratios (ORs) and 95% confidence intervals (95% CI). The two models included (1) minimally adjusted: age and baseline pain; (2) fully adjusted: age, baseline pain, BMI, smoking status, years in profession, level of leisure-time physical activity, number of daily patient transfers/handlings; and variables related to the psychosocial work environment (recognition and influence).

## 3. Results

Table 1 shows the baseline characteristics of the total sample (n = 1185). Nurses had a mean of 18.6 (SD: 11.4) years of experience and performed 3.8 (SD: 2.1) patient transfers per day. The mean intensity of pain was 2.37 (SD: 2.59), 0.38 (SD: 1.25), and 1.05 (SD: 1.91) for the shoulder, elbow, and hand/wrist regions, respectively. The mean mental health score was 81.8 (SD: 13.6) and the mean vitality score was 64.1 (SD: 17.8). Table 2 shows the results of the cumulative logistic regression for the associations between vitality, mental health, and the odds of a one-point increase in upper extremity pain intensity during follow-up.

**Table 1.** Baseline characteristics of the total sample.

| Characteristics | Value |
|---|---|
| **Age (years)** | 48.4 (10.4) |
| **BMI (kg/m$^2$)** | 25.2 (4.7) |
| **Pain intensity (0–10)** | |
| Shoulder | 2.3 (2.59) |
| Elbow | 0.38 (1.25) |
| Hand/wrist | 1.05 (1.91) |
| **Smoke status** | |
| Yes | 8% |
| No | 92% |
| **Leisure-time physical activity** | |
| Sedentary | 5.3% |
| Light exercise | 64.7% |
| Moderate | 28.1% |
| Vigorous | 1.9% |
| **Years in profession** | 18.6 (11.4) |
| **Frequency of daily patient transfers** | 3.8 (2.1) |
| **Vitality (SF-36)** | |
| Low (0–50) | 28.3% |
| Moderate (>50–75) | 47.2% |
| High (>75–100) | 24.6% |
| Total score (0–100) | 64.06 (17.8) |

**Table 1.** *Cont.*

| Characteristics | Value |
|---|---|
| **Mental health (SF-36)** | |
| Low (0–50) | 4.8% |
| Moderate (>50–75) | 26.1% |
| High (>75–100) | 69.1% |
| Total score (0–100) | 81.8 (13.6) |
| **Psychosocial work environment** | |
| Recognition from colleagues | 79 (15.9) |
| Influence at work | 74.0 (17.0) |

Data are presented as mean (standard deviation) or percentages (%).

**Table 2.** Cumulative logistic regression for mental health, vitality, and odds of one-point increase in upper limb pain intensity (0–10) during 12 months of follow-up.

| Exposure | Pain Region | SF-36 Score | Model 1 [a] OR (95%CI) | Model 2 [b] OR (95%CI) |
|---|---|---|---|---|
| Vitality | Shoulder | High | ref | ref |
| | | Mod | **1.47 (1.14–1.90)** | **1.35 (1.05–1.75)** |
| | | Low | **2.24 (1.68–3.00)** | **1.96 (1.43–2.68)** |
| | Elbow | High | ref | ref |
| | | Mod | 1.36 (0.83–2.24) | 1.24 (0.73–2.11) |
| | | Low | **1.78 (1.01–3.14)** | 1.49 (0.81–2.74) |
| | Hands/wrist | High | ref | ref |
| | | Mod | **2.03 (1.44–2.86)** | **2.09 (1.46–2.98)** |
| | | Low | **2.25 (1.56–3.25)** | **2.32 (1.58–3.42)** |
| Mental Health | Shoulder | High | ref | ref |
| | | Mod | **1.57 (1.22–2.00)** | **1.50 (1.16–1.93)** |
| | | Low | 1.56 (0.90–2.72) | 1.37 (0.78–2.41) |
| | Elbow | High | ref | ref |
| | | Mod | 1.25 (0.87–1.80) | 1.20 (0.83–1.75) |
| | | Low | 1.55 (0.68–3.50) | 1.32 (0.61–2.86) |
| | Hands/wrist | High | ref | ref |
| | | Mod | 1.05 (0.80–1.37) | 1.02 (0.78–1.35) |
| | | Low | 1.40 (0.76–2.58) | 1.36 (0.72–2.56) |

Values are presented as odds ratios and 95% confidence intervals; ref = reference category; significant associations are shown in bold. [a] Minimally adjusted: age and baseline pain; [b] Fully adjusted: age, baseline pain, body mass index, smoking status, years in profession, level of leisure-time physical activity, number of daily patient visits, and psychological variables (recognition and influence).

Using "high vitality" as reference, in the fully adjusted model, significative associations between low vitality levels and the odds of increased shoulder pain (OR = 1.96; 95% CI: 1.43–2.68) and hand/wrist pain (OR = 2.32; 95% CI: 1.58–3.42) was observed. These results were consistent with the crude model. Similarly, significant associations were observed between moderate levels of vitality and the odds of increased shoulder pain (OR = 1.35; 95% CI: 1.05–1.75) and hand/wrist pain (OR = 2.09; 95% CI: 1.46–2.98). These results were consistent with the crude model. On the other hand, no association was observed between vitality and pain intensity in the elbow region in the adjusted full model, but in the crude model a significant association was found between low levels of vitality and the odds of increased elbow pain (OR = 1.78, 95% CI 1.01–3.14).

Using "high level of mental health" as reference, a significant association was observed in the fully adjusted model between moderate levels of mental health and shoulder pain intensity (OR = 1.50, 95% CI: 1.16–1.93), which was consistent with the results of the crude model (OR = 1.57, 95% CI: 1.22–2.0). No association was observed between mental health and pain intensity in the elbow and hand/wrist regions.

## 4. Discussion

This prospective study shows significant associations between levels of vitality and odds of increased pain intensity in the shoulder and hand/wrist regions among female hospital nurses. These findings represent new and relevant knowledge for this group of healthcare professionals, which could help guide nursing managers in decision making and in the development of future preventive strategies aimed at maintaining high vitality among hospital nurses and reducing their exposure to stressful/extenuating work situations. In particular, hospital managers should seek to address controllable work-related stressors and support nurses' emotional competence [36].

Vitality had a stronger association with increased pain intensity during follow-up among hospital nurses compared with mental health, which only showed an association between moderate levels of mental health and shoulder pain. In line with our results, Hiestand et al. found that excessive fatigue is associated with pain, sleep, and mental health in Norwegian nurses [37]. Vitality is an important indicator associated with a worker's daily well-being and employee performance [21,22]. Thus, greater vitality in nurses could be a strength to better cope with work demands in a healthcare setting, in addition to having personal or occupational resources such as self-efficacy and optimism [38]. In this scenario, higher levels of vitality may be a protective factor against work stress and thus mitigate increased pain in the shoulder and hand/wrist region [39]. High vitality also correlates with higher emotional and social functioning and lower perceived stress [23], which are considered one of the key factors associated with the prevalence of upper extremity musculoskeletal disorders among nurses [19]. Furthermore, consistent with our findings, a prospective cohort study of community-based women found that lower levels of vitality are also significantly associated with persistent, high-intensity low back pain and disability [40]. Indeed, vitality is associated with mortality in middle-aged people, which further underlines its relevance [41].

With regard to mental health, two previous systematic reviews recommended that clinicians assess patients' psychological status (e.g., depression) in people with upper limb disorders at risk of work absenteeism [26,42]. Mental health problems have a significant impact on the occurrence of musculoskeletal pain among nurses in teaching hospitals [43]. Regarding the lack of association between mental health and hand/wrist pain intensity and the weaker associations, this result could be explained by the overall high scores provided in our sample, representing better mental health. Despite this, we observed a significant association between moderate levels of mental health and the likelihood of increased shoulder pain at follow-up. Vitality and mental health have been shown to be positively associated in women with persistent pain [23]. Therefore, both variables should be constantly monitored among nurses in the hospital setting.

We found that in hospital nurses, pain intensity was greater in the shoulder region compared to the elbow or hand/wrist region at baseline and during follow-up. Along these lines, several scientific reports have indicated that the shoulder is one of the most affected sites among hospital nurses [1,7,12]. The incidence of shoulder pain is commonly higher in nurses with a high level of work-related stress [44], and non-specific shoulder pain may be related to burnout and depressive symptoms [45]. On the other hand, repetitive wrist or finger movements, as well as high work stress and mental workload, may also influence a higher prevalence of pain in the hand/wrist region [13]. On the other hand, it is important to note that both mental health and vitality were not associated with pain intensity in the elbow region in the fully adjusted model. This could be explained by the fact that the intensity of elbow pain was low in this sample. Overall, elbow pain is the

least frequent of the musculoskeletal disorders among hospital nurses [46]. Furthermore, despite the low pain intensity reported in the present study, a previous study indicated that pain levels $\leq 5$ slightly interfere with the functioning [34]. It may also be relevant to treating mild pain in order to avoid the future chronification or aggravation of pain.

Finally, these results reinforce the need for hospital nursing unit managers to focus on both physical and psychosocial risks in order to promote better physical and mental health among workers, considering vitality as a crucial factor as well. Our results highlight the need to design intervention programs to reduce musculoskeletal symptoms in hospital nurses and to improve their vitality and mental health. For example, physical therapy combined with psychological intervention (based on cognitive behavioral therapy concepts) for women suffering from chronic pelvic pain has been shown to be effective in reducing perceived stress and increasing vitality [23]. Also, educational interventions that provide specialized information that improves women's control over the experience of a particular health condition have been shown to increase vitality with long-term effects [47,48].

The well-being of healthcare personnel is key to the quality of hospital services. Therefore, future research should continue to address this issue in order to improve management capacity and prevent musculoskeletal conditions among nursing staff. In this context, other psycho-emotional variables that could affect both the physical and mental health of healthcare personnel should also be taken into account. For example, in a context of a social crisis (e.g., the COVID-19 pandemic), emotional intelligence and resilience may be crucial for coping with adversity [49]. Pain catastrophizing and symptoms of anxiety and depression are also associated with absenteeism from work in people with upper extremity disorders [26]; therefore, they should be taken into account in return-to-work programs. On the other hand, burnout in healthcare workers and general well-being are closely related [50]. Thus, burnout can lead to negative emotions [51]. In the occupational setting, a higher proportion of positive emotions relative to negative emotions leads to greater work engagement (i.e., dedication and vigor) and, consequently, higher levels of subjective well-being [52]. This suggests the importance of addressing positive emotions in prevention and intervention programs for nurses. Finally, decision makers should focus on the psychological intervention needed for those nurses who are most vulnerable (e.g., those on temporary contracts and/or without family support). First and foremost, managers should distribute tasks appropriately to avoid psychological distress caused by overwork [53]. From a primary prevention perspective, it is necessary to promote work environments that ensure that work characteristics do not have a negative impact on mental health or well-being.

*Strengths and Limitations*

The main strengths of this study consist of the relatively large sample size, the prospective design with 12 months of follow-up, the use of validated scales to assess predictors and outcome, and the statistical adjustments for several confounding factors (demographic, physical, and psychosocial). Limitations include recall bias and the possibility of underreporting musculoskeletal disorders, which would increase the possibility of misclassification bias and underestimate the true size of the association. Likewise, there remains a possibility of residual confounding that could attenuate the observed association. Also, the meaning of work and role conflict were not considered as confounding variables and future research should take this into consideration. Another limitation was that the correlations of other important variables collected, such as physical activity, years in the profession, recognition, and influences, were not studied. However, this was not the aim of the study, and these variables were only used as co-variables in the regression model. Finally, only nurses from hospital settings were included, which could influence the generalizability of the results.

## 5. Conclusions

Female hospital nurses with low/moderate vitality and moderate mental health were more likely to have increased upper extremity pain intensity at 1-year follow-up compared with nurses with high vitality and good mental health. These findings provide an important incentive for nurse managers to assess the vitality of hospital nurses and include this factor in prevention strategies to ensure a healthy workforce and, in turn, high quality care.

**Author Contributions:** Conceptualization, R.N.-C., A.E. and L.L.A.; methodology, J.V., M.D.J. and L.L.A.; software, L.L.A.; validation, L.L.A.; formal analysis, L.L.A.; investigation, R.N.-C., A.E. and L.L.A.; resources, L.L.A.; data curation, L.L.A.; writing—original draft preparation, R.N.-C.; writing—review and editing, R.N.-C., A.E., J.C., S.P.-A., R.L.-B., C.C.-M., J.V., M.D.J. and L.L.A.; visualization, L.L.A.; supervision, L.L.A.; project administration, L.L.A.; funding acquisition, L.L.A. All authors have read and agreed to the published version of the manuscript.

**Funding:** Author L.L.A. obtained a grant from the Danish Working Environment Research Fund (Arbejdsmiljøforskningsfonden) for this study. Grant number 26-2015-09. Author R.N.-C. is supported by the National Research and Development Agency of Chile (ANID/2020-72210026). Author R.L.-B. is supported by the European Union—Next Generation EU.

**Institutional Review Board Statement:** The National Research Centre for the Working Environment has an agreement with the Danish Data Protection Agency to register all studies in-house. According to Danish legislation, studies based on research questionnaires and registers do not require approval from ethical and scientific committees or informed consent (Act on Scientific Ethical Committees, §14.2).

**Informed Consent Statement:** According to Danish law (Scientific Ethical Committees Act, §14.2), scientific studies based on questionnaires do not require the informed consent of study participants. All data were de-identified and analyzed anonymously.

**Data Availability Statement:** The data used to support the findings of current study are available from author L.L.A. upon request.

**Conflicts of Interest:** The authors declare no conflict of interest.

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
