# Peer review of "Can Vitality and Mental Health Influence Upper Extremity Pain? A Prospective Cohort Study of 1185 Female Hospital Nurses"

_ejihpe, doi:10.3390/ejihpe13100154_

Round 1

Reviewer 1 Report

I do not agree when the authors indicate that scientific studies based on questionnaires do not require the approval of official ethical or scientific committees or the informed consent of the study participants.

Author Response

We thank the Reviewer #1 for this valuable comment. In the new version of our manuscript, we have been more specific on this point, detailing the following: “The National Research Centre for the Working Environment has an agreement with the Danish Data Protection Agency to register all studies in-house. According to Danish legislation, studies based on research questionnaires and registers do not require approval from ethical and scientific committees or informed consent (Act on Scientific Ethical Committees, §14.2)”.

Also, as suggested by the editor, we provide as a citation the following website that supports this statement.

https://nationaltcenterforetik.dk/ansoegerguide/overblik/hvad-skal-jeg-anmelde

Reviewer 2 Report

I am grateful for the opportunity to review this interesting article and I congratulate the authors for chosing this issue.

I present suggestions and questions for improvement in attached document.

Only 25% of references are within the last 5 years...

Regards,

Author Response

We are grateful for the comments of Reviewer #2, which have contributed to significantly improve the quality of the manuscript. We have uploaded a new version of the manuscript (with marked changes). We have also added new and more recent references (last 5 years) in both the introduction and discussion of our manuscript. We hope that the revised version of our manuscript has met the expectations of Reviewer #2.

Reviewer 3 Report

Dear Authors,

The objective of this study was to examine the prospective association between upper extremities pain and vitality levels, using data from a prospective cohort of nurses from 19 Danish hospitals. The objective of the study is clear, and the methods appear sound; I do have some comments though.

My main concern is about the practical significance of the results and their implications in terms of prevention at the workplace level.

According to the authors, in one hand, UEMSD pain is associated with work-related psychosocial factors; in the other hand, vitality is associated with employee performance and well-being. Thus linking UEMSD pain and vitality level -as an individual positive factor of the perception of working situation-  could be of interest for the prevention. I can admit that, but to what extent can managers propose actions in the workplace to ensure healthy workforce? Could the authors be more specific about the practical implication of the results? In particular in terms of primary prevention.

Introduction

L76-77: please justify with references

Predictor variables

L112-113: Why did you classified the vitality and mental health scales into 3 categories? What was the rational for the chosen threshold? Not statistical as it can be seen with the distribution of mental scores.

Main outcome

L116: What is the clinical meaning for one point increase of pain?

Discussion

L208 : I agree that work-related physical and psychosocial risks need to be taken into account to ensure the physical and mental health of workers, but speaking about primary prevention, I'm not convinced that vitality needs to be as well.

Author Response

R: We thank Reviewer #3 for his valuable comments. Following the reviewers' recommendations We have added more current references that allow us to expand the introduction and discussion of our manuscript, emphasizing the relevance of our findings and the implications for clinical practice. Likewise, we have been more cautious with the interpretation of the results and have added some examples of how managers can propose actions in the workplace to ensure nurses' health. We have uploaded a new version of the manuscript (with marked changes). We hope that the revised version of our manuscript has met the expectations of Reviewer #3.

Introduction

L76-77: please justify with references

R: We have added a reference in this sentence

Predictor variables

L112-113: Why did you classified the vitality and mental health scales into 3 categories? What was the rational for the chosen threshold? Not statistical as it can be seen with the distribution of mental scores.

R: We thank the reviewer for this valuable comment. We have added the following sentence to clarify this point:These cut-off points were taken into account considering that the components of the SF-36 questionnaire are standardised to reflect a mean of 50 in the general population. Whereas scores above 75 have been described for the young population in the European population, weighted by age and sex [32].”

Main outcome

L116: What is the clinical meaning for one point increase of pain?

R: We thank the reviewer for this valuable comment. We have added the following sentence to highlight what is the clinical significance of a single point increase in pain: “We chose an increase of one point on the pain scale because this is the smallest possible increase on this ordinal scale. It was also taken into account that the baseline pain intensity in the sample was low. In this sense, an increase of one point could be relevant, as a previous study showed that even low levels of pain affect functioning in people with chronic pain [33].”

This may therefore be relevant to the prevention of work-related musculoskeletal pain in nurses

Discussion

L208 : I agree that work-related physical and psychosocial risks need to be taken into account to ensure the physical and mental health of workers, but speaking about primary prevention, I'm not convinced that vitality needs to be as well.

R: We thank the reviewer for this valuable comment. We have added a paragraph at the end of the discussion about the relevance of evaluating other psychoemotional factors in future research, which could also be relevant in health care workers.

Reviewer 4 Report

Thank you for the opportunity to review this paper. The manuscript entitled "Can Vitality and Mental Health Influence Upper Extremity Pain? A Prospective Cohort Study of 1,185 Hospital Nurses" aims to analyze the prospective association between vitality mental health, and upper extremity pain intensity in female hospital nurses. 

In my opinion, the manuscript is not yet ready for publication. Many parts need to be reviewed, such as those listed below.

Abstract

1) The abstract is missing important data such as the gender percentage of the sample, and the range, mean, and standard deviation of age. These details should be included in the abstract.

Introduction

2) The introduction as it is is excessively short. In this part, it is necessary to deepen the starting literature with a stronger and more recent theoretical framework from the point of view of literature. I believe it can be expanded with a more specific overview of the factors involved and more recent literature that highlights the issue. It is also useful to introduce not only national but also international literature, specifically with similar countries of the study (Italy, France, Portugal, etc.).

Method

3) On line 91 to line 100 of the Method section there is a lot of information that should go in the Procedure section. Furthermore, the number of participants, the mean and standard deviation of age, and the percentage of gender are not indicated. These data must be indicated in this section.

4) Regarding the Short Form Health Survey (SF-36), we need to be clearer when explaining the scores. The scale has 6 points, but from 0 to 5 or from 1 to 6? What do "All the time" and "Not at all" correspond to? It is also unclear how scores are calculated.

5) Finally, in the sentence "from 0 to 100 (higher score is better)", the term "better" is not acceptable in a scientific journal: it can be said that higher scores correspond to higher levels of well-being.

6) I find no mention of the study being approved by an ethics committee. For this, I believe it is necessary to consult your Internal Review Board (IRB) for approval, which will be requested by the journal.

Results

7) Lines 137 to 139: This information (as mentioned above) should be reported in the Participant’s section.

8) The results paragraph is very meager: it is necessary to describe the results shown in the table, illustrating their significance, bootstrap, etc.

9) Furthermore, in the title of Table 2 it must be indicated that it is a regression model: everything must be very clear.

Discussion

10) As with the introduction section, I think the discussions are too short. I believe it must be expanded. Here are some recent works that suit your theme and which I think may be useful for expanding and updating the introduction section:

- Angelini, G., Buonomo, I., Benevene, P., Consiglio, P., Romano, L., & Fiorilli, C. (2021). The Burnout Assessment Tool (BAT): A contribution to Italian validation with teachers’. Sustainability13(16), 9065.

- Buonomo, I., Santoro, P. E., Benevene, P., Borrelli, I., Angelini, G., Fiorilli, C., ... & Moscato, U. (2022). Buffering the Effects of Burnout on Healthcare Professionals’ Health—The Mediating Role of Compassionate Relationships at Work in the COVID Era. International Journal of Environmental Research and Public Health19(15), 8966.

- Fiorilli, C., Pepe, A., Buonomo, I., & Albanese, O. (2017). At-risk teachers: the association between burnout levels and emotional appraisal processes. The Open Psychology Journal10(1).

- López-Angulo, Y., Mella-Norambuena, J., Sáez-Delgado, F., Peñuelas, S. A. P., & González, O. U. R. (2022). Association between teachers’ resilience and emotional intelligence during the COVID-19 outbreak. Revista Latinoamericana de Psicología54, 51-59.

- Pluskota, M., & Zdziarski, K. (2022). Mental resilience and professional burnout among teachers. Journal of Education, Health and Sport12(3), 249-267.

- Pyhältö, K., Pietarinen, J., Haverinen, K., Tikkanen, L., & Soini, T. (2020). Teacher burnout profiles and proactive strategies. European Journal of Psychology of Education, 1-24.

- Rusu, P. P., & Colomeischi, A. A. (2020). Positivity ratio and well-being among teachers. The mediating role of work engagement. Frontiers in psychology11, 1608.

However, I invite you to deepen the literature on the subject to have a more articulated and complex overview of the variables studied.

11) On line 223 there is a typo: "association.Likewise"

References

12) The references section reflects the scant literature included in the study, both in the introduction and discussion sections. The theoretical panorama must be complete, for this reason, 32 references are not enough for an entire article. Furthermore, almost half of the references are before 2013 and are too much earlier than the transformations and scientific updates of the literature of the last ten years. Reference should be made to a wider and more up-to-date literature.

Author Response

Abstract

  • The abstract is missing important data such as the gender percentage of the sample, and the range, mean, and standard deviation of age. These details should be included in the abstract.

R: We have included the mean and standard deviation of age in the abstract. In addition, we have further emphasised that only women were included in this study (both in the abstract and in the title).

Introduction

2) The introduction as it is is excessively short. In this part, it is necessary to deepen the starting literature with a stronger and more recent theoretical framework from the point of view of literature. I believe it can be expanded with a more specific overview of the factors involved and more recent literature that highlights the issue. It is also useful to introduce not only national but also international literature, specifically with similar countries of the study (Italy, France, Portugal, etc.).

R: We agree with the Reviewer on this point. Following his recommendations, we have further developed the background literature with a stronger and more recent theoretical framework in terms of literature, including also international literature from countries similar to those of the study.

Method

3) On line 91 to line 100 of the Method section there is a lot of information that should go in the Procedure section. Furthermore, the number of participants, the mean and standard deviation of age, and the percentage of gender are not indicated. These data must be indicated in this section.

R: The suggested changes have been added. The information presented in the results has been moved to the section "Study design and participants". In addition, we have further emphasised that only women were included in this study (both in the abstract and in the title).

4) Regarding the Short Form Health Survey (SF-36), we need to be clearer when explaining the scores. The scale has 6 points, but from 0 to 5 or from 1 to 6? What do "All the time" and "Not at all" correspond to? It is also unclear how scores are calculated.

R: We thank Reviewer #4 for this valuable comment. We agree that this point should be presented more clearly and have made the appropriate corrections.

5) Finally, in the sentence "from 0 to 100 (higher score is better)", the term "better" is not acceptable in a scientific journal: it can be said that higher scores correspond to higher levels of well-being.

R: We have modified this sentence, stating specifically those mentioned in the original questionnaire.

6) I find no mention of the study being approved by an ethics committee. For this, I believe it is necessary to consult your Internal Review Board (IRB) for approval, which will be requested by the journal.

R: We thank the reviewer for this valuable comment. In the new version of our manuscript, we have been more specific on this point, detailing the following: “The National Research Centre for the Working Environment has an agreement with the Danish Data Protection Agency to register all studies in-house. According to Danish legislation, studies based on research questionnaires and registers do not require approval from ethical and scientific committees or informed consent (Act on Scientific Ethical Committees, §14.2)”.

Also, as suggested by the editor, we provide as a citation the following website that supports this statement.

https://nationaltcenterforetik.dk/ansoegerguide/overblik/hvad-skal-jeg-anmelde

Results

7) Lines 137 to 139: This information (as mentioned above) should be reported in the Participant’s section.

R: We thank reviewer #4 for this suggestion.The above information has been moved to the "Study design and participants" section.

8) The results paragraph is very meager: it is necessary to describe the results shown in the table, illustrating their significance, bootstrap, etc.

R: We are grateful for this comment. We have provided more details in the results section, to better describe the information available in the tables.

9) Furthermore, in the title of Table 2 it must be indicated that it is a regression model: everything must be very clear.

R:This information has been added to the title of table 2.

Discussion

10) As with the introduction section, I think the discussions are too short. I believe it must be expanded. Here are some recent works that suit your theme and which I think may be useful for expanding and updating the introduction section:

- Angelini, G., Buonomo, I., Benevene, P., Consiglio, P., Romano, L., & Fiorilli, C. (2021). The Burnout Assessment Tool (BAT): A contribution to Italian validation with teachers’. Sustainability13(16), 9065.

- Buonomo, I., Santoro, P. E., Benevene, P., Borrelli, I., Angelini, G., Fiorilli, C., ... & Moscato, U. (2022). Buffering the Effects of Burnout on Healthcare Professionals’ Health—The Mediating Role of Compassionate Relationships at Work in the COVID Era. International Journal of Environmental Research and Public Health19(15), 8966.

- Fiorilli, C., Pepe, A., Buonomo, I., & Albanese, O. (2017). At-risk teachers: the association between burnout levels and emotional appraisal processes. The Open Psychology Journal10(1).

- López-Angulo, Y., Mella-Norambuena, J., Sáez-Delgado, F., Peñuelas, S. A. P., & González, O. U. R. (2022). Association between teachers’ resilience and emotional intelligence during the COVID-19 outbreak. Revista Latinoamericana de Psicología54, 51-59.

- Pluskota, M., & Zdziarski, K. (2022). Mental resilience and professional burnout among teachers. Journal of Education, Health and Sport12(3), 249-267.

- Pyhältö, K., Pietarinen, J., Haverinen, K., Tikkanen, L., & Soini, T. (2020). Teacher burnout profiles and proactive strategies. European Journal of Psychology of Education, 1-24.

- Rusu, P. P., & Colomeischi, A. A. (2020). Positivity ratio and well-being among teachers. The mediating role of work engagement. Frontiers in psychology11, 1608.

However, I invite you to deepen the literature on the subject to have a more articulated and complex overview of the variables studied.

R: We thank the reviewer for this important comment. We have expanded the discussion section, as well as the introduction, taking into account some recent recommended references that are more relevant to our topic.

11) On line 223 there is a typo: "association.Likewise"

R: We are grateful for this observation This error has been corrected

References

12) The references section reflects the scant literature included in the study, both in the introduction and discussion sections. The theoretical panorama must be complete, for this reason, 32 references are not enough for an entire article. Furthermore, almost half of the references are before 2013 and are too much earlier than the transformations and scientific updates of the literature of the last ten years. Reference should be made to a wider and more up-to-date literature.

R: We thank the reviewer for this important comment. We have added more than 15 current references (i.e. published in the last 5 years) that allow us to expand the introduction and discussion of our manuscript, emphasising the relevance of our findings and the implications for clinical practice.

Round 2

Reviewer 3 Report

Dear Authors,

Thank you for the greatly improved revised version of your manuscript. I have a few additional questions/comments.

About covariates (L261):

Why did you choose only recognition and influence at work as adjustment factors? I'm thinking in particular of the meaning of work and role conflict.

About discussion (L308-309)

I believe that the ability of an organization to create a "healthy" work climate is crucial for the well-being of nurses. From a primary prevention perspective, it is necessary to promote work environments that ensure that the objective characteristics of work (as well as the resulting perceived working situations) do not have a negative impact on mental health or well-being. The last sentence of the paragraph (L308-309) is not sufficient to reinforce this need. I therefore suggest replacing “at the same time” with “First and foremost”.

Corrections:

The words "musculoskeletal" (L43) and "at" (L128) appear twice.

L277: The beginning of the sentence is in bold.

Author Response

Authors' response: First of all, we thank reviewer #3 for his valuable comments, which contribute considerably to improving the quality of our manuscript.

About covariates (L261): 

Why did you choose only recognition and influence at work as adjustment factors? I'm thinking in particular of the meaning of work and role conflict.

Authors' response: These variables were chosen primarily because we only included female nurses from nursing departments where physical transfer of patients was performed (and excluded administrative and clerical departments), so we felt that role conflict might be more homogeneous among nurses. However, we agree that these are relevant variables and have stated this as a limitation (Line 322-323).

About discussion (L308-309)

I believe that the ability of an organization to create a "healthy" work climate is crucial for the well-being of nurses. From a primary prevention perspective, it is necessary to promote work environments that ensure that the objective characteristics of work (as well as the resulting perceived working situations) do not have a negative impact on mental health or well-being. The last sentence of the paragraph (L308-309) is not sufficient to reinforce this need. I therefore suggest replacing “at the same time” with “First and foremost”.

Authors' response: We thank Reviewer #3 for this valuable comment. We agree that this point should be reinforced. Therefore, we have made the suggested change and also added the following sentence to provide a clearer message to the reader:

“From a primary prevention perspective, it is necessary to promote work environments that ensure that work characteristics do not have a negative impact on mental health or well-being.” (Line 310-312).

Corrections:

The words "musculoskeletal" (L43) and "at" (L128) appear twice.

L277: The beginning of the sentence is in bold.

Authors' response:We thank the reviewer for his comments. These errors have been corrected

Reviewer 4 Report

Thank you for taking my advice. I have reviewed the changes made and I am pleased to see that the manuscript has improved significantly. I appreciate the effort you put into addressing my comments and suggestions. Your revisions have made the manuscript more clear. Thank you

Author Response

We thank reviewer nº 4 for his valuable comments; his appreciations were very relevant to improve the quality of our manuscript.